# ResPilot: Teleoperated Finger Gaiting via Gaussian Process Residual Learning

**Patrick Naughton[1,2*], Jinda Cui[2], Karankumar Patel[2], Soshi Iba[2]**
[1]UIUC, [2]Honda Research Institute USA
pn10@illinois.edu, {jinda_cui, karankumar_patel, siba}@honda-ri.com
*This work was completed during an internship at Honda Research Institute USA

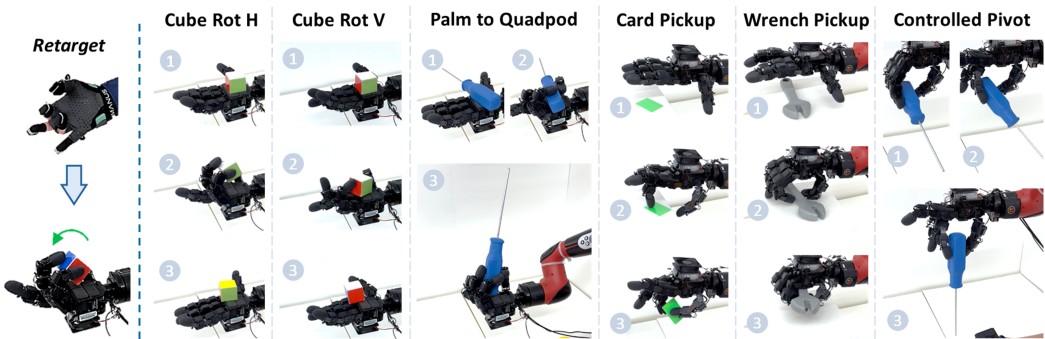

Figure 1: Our method retargets human hand configurations to a robot hand, enabling finger-gaited in-hand manipulation. We evaluate our method on six highly dexterous tasks with the palm facing upward and downward.

**Abstract:** Dexterous robot hand teleoperation allows for long-range transfer of human manipulation expertise, and could simultaneously provide a way for humans to teach these skills to robots. However, current methods struggle to reproduce the functional workspace of the human hand, often limiting them to simple grasping tasks. We present a novel method for finger-gaited manipulation with multi-fingered robot hands. Our method provides the operator enhanced flexibility in making contacts by expanding the reachable workspace of the robot hand through residual Gaussian Process learning. We also assist the operator in maintaining stable contacts with the object by allowing them to constrain fingertips of the hand to move in concert. Extensive quantitative evaluations show that our method significantly increases the reachable workspace of the robot hand and enables the completion of novel dexterous finger gaiting tasks. Project website: respilot-hri.github.io.

**Keywords:** Teleoperation, Dexterous Manipulation, Gaussian Process

## 1 Introduction

Teleoperated dexterous manipulation has the potential to enable the long-range transfer of human manipulation skills to remote environments and could simultaneously serve as a mass data collection mechanism to enable fully autonomous manipulation. Fluent manipulation using a multi-fingered robot hand is difficult even with a human operator in the loop because it requires precise control of the many degrees of freedom (DoFs) of the hand simultaneously to coordinate finger motion. In particular, finger-gaited manipulation, where a subset of fingers are used to maintain a contact state while other fingers move to change their contact state with the object and move it, has proven challenging because the motion of the target object is sensitive to the locations and modes of con-

8th Conference on Robot Learning (CoRL 2024), Munich, Germany.

tacts. Such challenges are exacerbated by limitations of current interfaces such as a lack of haptic feedback and kinematic mismatches between the human and robot.

Existing approaches for teleoperation of dexterous hands typically use a retargeter to map the operator's hand to the robot's desired configuration. Current literature often evaluates these retargeters by how well they allow the robot hand to match the operator's hand visually, or to achieve grasping and pushing tasks [1, 2, 3, 4]. A few works have attempted basic in-hand manipulation [5, 6] but tend to require specially designed task spaces and provide no details about their reliability or efficiency. To the author's knowledge, Handa et al. [7] is the only existing method that has demonstrated teleoperated finger gaiting with a multi-fingered hand, and even this method was only able to achieve a single finger gaiting task (in-hand block rotation).

The main contribution of this work is a novel retargeting method that enables previously unseen teleoperated finger gaiting. We address two core challenges: 1) making contacts using free fingers at diverse locations on the target object and 2) using these contacts to stably change the object's pose. Our method uses a small set of hand-labeled calibration poses to learn a residual Gaussian Process between an optimization-based retargeting method and the labeled robot configurations. We also allow the operator to constrain the distance between the robot's fingertips while still tracking their input motion to allow the robot to maintain stable contact with a grasped target object as the fingers move. We show that this method is fast to calibrate, expands the reachable workspace of the robot, and ultimately enables previously unseen teleoperated finger gaiting.

## 2 Related Work

There are many existing approaches for retargeting a person's hand motion to an anthropomorphic robot hand. Here, we review the ones most related to our method. For a more comprehensive treatment, we refer readers to [8].

Joint-space retargeters directly map each joint of the operator's hand to a joint of the robot hand and command the robot joints to follow the operator's hand [9, 10, 11]. While this approach allows the robot's fingers to approximate the shapes of the operator's, it tends to make precise fingertip control difficult due to kinematic differences between the robot and operator. Conversely, inverse kinematics (IK) retargeters [12, 13], which use IK to directly command each robot fingertip to match the pose of the operator's fingertips, enable precise fingertip grasping, but can result in unintuitive finger shapes [14, 9]. Additionally, since robot hands are often larger than the operator's, this approach can prevent the operator from reaching much of the robot's workspace. In contrast, our method, by calibrating for important configurations at multiple points in the robot's workspace, enables our retargeter to enjoy the advantages of both methods.

Recently, several systems [7, 2, 4] have used a hand keypoint-vector matching (HKVM) approach where corresponding keypoints are labeled on the operator and robot hands (for example, the fingertips and/or palm). A set of pairs of keypoints is chosen to define vectors on the robot and operator hands and the desired robot configuration is computed as the one that minimizes the deviation between these vectors [7]. While Handa et al. [7] demonstrated some basic finger gaiting using this method, their focus on fingertip grasping typically limits them to grasping tasks. In contrast, we show that our retargeter enables a suite of highly dexterous finger gaiting tasks.

Finally, a few works have proposed "pure-learning" approaches to the retargeting problem where the function from operator hand configuration to robot hand configuration is directly learned from a set of labeled examples [15, 1]. While these methods are capable of both power and precision grasping, to the author's knowledge researchers have not used these methods for finger gaiting.

## 3 Method

The goal of a retargeting method is to map a given operator configuration to a commanded robot configuration. Our method uses a small set of calibration poses to learn a residual between an

optimization-based retargeter and the labeled robot poses conditioned on the operator's hand configuration. We assume access to measurements of the operator's fingertip poses and joint angles.

## 3.1 Optimization-based Retargeting Methods

For a given configuration of the operator's hand, existing optimization-based methods produce $\mathbf{q}_o^*$ by minimizing the error between the $H$ sensed keypoint vectors on the operator's hand and the scaled vectors between the same keypoints on the robot hand [7, 2, 4]:

$$\mathbf{q}_o^*(\mathbf{q}_h) = \arg\min_{\mathbf{q}_o} \sum_{i=1}^{H} \|\mathbf{r}_i(\mathbf{q}_o) - \beta \mathbf{h}_i(\mathbf{q}_h)\|^2 + \gamma \|\mathbf{q}_o\|^2 \tag{1}$$

where $\mathbf{r}_i(\cdot)$ and $\mathbf{h}_i(\cdot)$ compute the $i$th keypoint vector for the robot and human hands, $\mathbf{q}_o$ and $\mathbf{q}_h$ denote the robot and human hand configurations, and $\beta$ is a scaling parameter to account for size differences between the hands. $\gamma$ is a regularization parameter that biases the robot configuration to be close to zero (an open hand). Following Handa et al. [7], we use the same 10 keypoint vectors shown in Figure 2 and set $\beta = 1.6$, $\gamma = 0.0025$. We refer to this retargeter as the "hand keypoint vector matching" (HKVM) retargeter since it attempts to match vectors between keypoints on the robot's hand to corresponding vectors on the operator's hand.

## 3.2 Residual Gaussian Process

While the HKVM retargeter alone can accomplish some tasks, it struggles to reach parts of the fingers' workspace that are vital for finger gaiting, such as near-palm grasps. To expand the reachable workspace, we use HKVM as a "base retargeter," and collect a small number $C$ of paired human hand and robot hand configurations, $D = \{(\mathbf{q}_{h_i}, \mathbf{q}_{r_i})\}_{i \in [C]}$, where $[C]$ denotes the sequence of integers $(1, \ldots, C)$. Let $F$ denote the shared set of fingers between the human and robot hands (in the case of a four-fingered robot hand used here, $F = \{\text{thumb, index, middle, ring}\}$) and denote by $\mathbf{q}[f]$ the

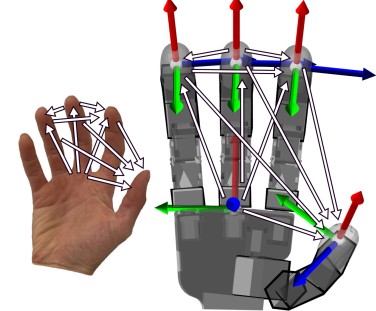

Figure 2: Hand keypoint vectors.

subset of hand joints associated with finger $f \in F$. Using $D$, we learn the hyperparameters of a multi-output Gaussian Process (GP) to regress the residual between $\mathbf{q}_o^*(\mathbf{q}_{h_i})$ and $\mathbf{q}_{r_i}$ from $\mathbf{q}_{h_i}$. We denote this full residual with $\boldsymbol{\xi}(\mathbf{q}_h)$. In practice, we assume that the desired residuals of each finger are independent of each other and that they are only functions of the configuration of the corresponding human finger. Thus, we learn a separate residual GP for each robot finger which takes as input only the joint angles of the corresponding human finger, $\boldsymbol{\xi}^f(\mathbf{q}_h[f])$. These assumptions make control of each finger more independent and thus easier for the operator to reason about, and reduce the dimensionality of the learning problem. They also enable partial labelling of pairs of hand configurations, as was done by Correia Marques et al. [15]: for example, in fingertip pinching configurations, only the fingers involved in the pinch can be reliably labeled with a corresponding robot configuration. By only using these configurations to train the associated finger models, we keep the data for each finger model less noisy (for a given hand configuration, we refer to the fingers being labeled as "active fingers").

A GP represents a function, such as $\boldsymbol{\xi}^f$, as an indexed set of random variables with the property that any finite subset has a Gaussian distribution [16]. In this case, the index set is $\mathbf{q}_h[f]$. The GP is completely specified by its mean and covariance functions $\mu(\mathbf{q}_h[f])$ and $k(\mathbf{q}_h[f], \mathbf{q}_h'[f])$:

$$\boldsymbol{\xi}^f(\mathbf{q}_h[f]) \sim GP(\mu(\mathbf{q}_h[f]), k(\mathbf{q}_h[f], \mathbf{q}_h'[f])) \tag{2}$$

In our case, we assume that $\mu$ is uniformly zero (in expectation, we exactly follow the base retargeter). We assume a parametric form for $k$:

$$k(\mathbf{q}_h[f], \mathbf{q}_h'[f]) = \exp\left(-\sum_{i=1}^{|f|} \frac{\arccos\left(\mathbf{V}(\mathbf{q}_h[f]_i)\mathbf{V}(\mathbf{q}_h'[f]_i)^{\intercal}\right)^2}{2\ell_i^2}\right) \tag{3}$$

where, for a vector of $m$ angles $\mathbf{q}$, $\mathbf{V}(\mathbf{q})$ denotes the matrix

$$\mathbf{V}(\mathbf{q}) = \begin{bmatrix} \cos(q_1) & \sin(q_1) \\ \vdots & \vdots \\ \cos(q_m) & \sin(q_m) \end{bmatrix},$$

$|f|$ denotes the number of joints on finger $f$, and arccos is applied elementwise. We also use $\mathbf{V}$ to compute the residual $\boldsymbol{\xi}^f = \mathbf{V}(\mathbf{q}_r[f]) - \mathbf{V}(\mathbf{q}_o^*(\mathbf{q}_h)[f])$ to avoid the discontinuities encountered by learning directly in the angle space [17]. Thus, for each finger, we learn a $2|f|$-dimensional vector-valued function. Rather than learning each component independently, we learn a task-covariance matrix $\mathbf{K}_t$, where the element at row $i$ column $j$ is the covariance between components $i$ and $j$ of the output [18]. Assuming the calibration datapoints were labeled with Gaussian noise having variance $\sigma^2$, we compute the input covariance matrix as $\mathbf{K}_\sigma = \mathbf{K} + \sigma^2 I$, where $\mathbf{K}$ is computed by evaluating $k$ at every pair of collected human joint angles. The full covariance matrix is then $\mathbf{K}_q = \mathbf{K}_t \otimes \mathbf{K}_\sigma$, where $\otimes$ is the Kronecker product.

We use GPyTorch [19, 20] to implement the GPs. We collect a set of calibration configurations and compute the HKVM target robot configuration for each, saving an ordered pair $(\boldsymbol{\xi}_i[f], \mathbf{q}_{h_i}[f])$ for each active finger. We then fit a separate GP to each finger's dataset by optimizing the entries of $\mathbf{K}_t$, $(\ell_i)_{i \in [|f|]}$, and $\sigma$ (collected into parameter $\boldsymbol{\theta}$) to maximize the marginal log likelihood of the data:

$$\log(p(\mathbf{Q}_r | D, \boldsymbol{\theta})) = -\frac{1}{2} \mathbf{Q}_r^\mathsf{T} \mathbf{K}_q^{-1} \mathbf{Q}_r - \frac{1}{2} \log |\mathbf{K}_q| - \frac{n}{2} \log 2\pi. \tag{4}$$

$\mathbf{Q}_r$ is the stacked vector of all residual targets for a finger. For each calibration point, we vectorize the $|f| \times 2$ residual matrix and concatenate all of these vectors to form $\mathbf{Q}_r$. We optimize this loss with respect to $\boldsymbol{\theta}$ with the Adam [21] optimizer, using a learning rate of $\alpha = 0.01$ and running for $E = 3000$ epochs, which we found to be sufficient to achieve convergence of the loss.

When using the retargeter, we first compute $\mathbf{q}_o^*$ using Equation 1. Then, for each finger, we find the posterior mean $\boldsymbol{\xi}^{f^*}$ [16] of the residual distribution at the current finger joint configuration conditioned on the collected calibration poses for the current user. For each finger, we compute desired joint angles by converting the $|f| \times 2$ matrix, $\mathbf{V}(\mathbf{q}_o^*[f]) + \boldsymbol{\xi}^{f^*}$, into a $|f|$-dimensional vector, finding the angle each row makes with the $x$-axis. We then collect each of these finger joint targets into $\mathbf{q}_d$, the full desired configuration of the robot hand.

### 3.3 Finger Constraints

Using the Residual GP retargeter, the operator can now move the robot fingers in a large workspace. However, stably maintaining contact while moving the object can still be difficult because the operator may not be able to visualize the robot's target configuration, or reason about the forces this will apply. To alleviate this problem, we allow the operator to apply constraints on the motions of fingers to couple them together. Similar to Handa et al. [7], our system allows the operator to constrain any of the robot's fingertips to stay a hand-specified distance $d = 1$ cm away from the robot's thumb tip. We compute the final constrained robot configuration by solving

$$\begin{aligned} \mathbf{q}_c = \arg\min_{\mathbf{q}} & \|\mathbf{q} - \mathbf{q}_d\|_2^2 \\ \text{s.t.} \quad & \|\mathbf{r}_i(\mathbf{q})\| = d \quad \forall\, i \in R_c \end{aligned} \tag{5}$$

where $R_c$ denotes the set of constrained fingertip vectors. In our implementation, $R_c$ only ever contains vectors between one of the robot's fingertips and its thumb. The operator can toggle each of the vectors' inclusion in $R_c$ by tapping a corresponding foot pedal.

## 4 Implementation

To instantiate our method, we designed a set of calibration poses and tested the effectiveness of the retargeter when controlling a physical Allegro robot hand.

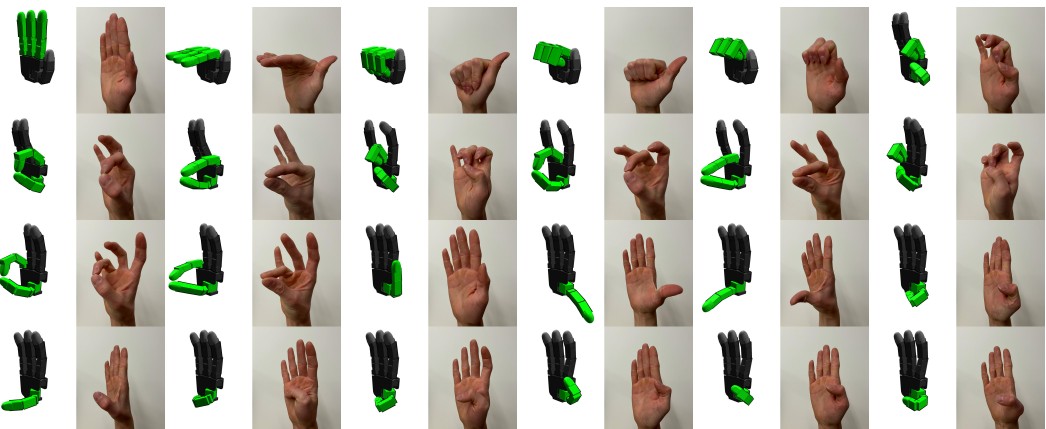

Figure 3: The full set of 24 calibration poses used to learn the residual GP. Active fingers for each calibration configuration are colored green.

**Calibration.** The goal of calibration is to expand the reachable workspace of the robot hand while maintaining an intuitive correspondence between the operator and robot hand shapes. We therefore selected calibration configurations at the boundary of the robot hand's workspace and labeled them using a human hand configuration at a corresponding boundary of its workspace. This heuristic can be broadly applied to other anthropomorphic hands as well. Figure 3 shows the full set of 24 calibration configurations, where the active fingers of each are colored green. We found that some pairs of configurations naturally defined a finger trajectory (e.g., moving from a pinch near the palm to far from the palm), and additionally recorded points halfway between these two extremes to constrain how the retargeter interpolated between them. After choosing a set of robot and human hand pairs, to calibrate the retargeter for a new operator, the operator simply matches their hand shape to each of the 24 configurations while their joint angles at each point are recorded. A GP for each finger is then trained to calibrate the retargeter.

**Inference.** Algorithm 1 shows how the final desired joint configuration is computed. To solve the mathematical programs on Lines 3 and 8 we use the NLopt library [22] with the SLSQP algorithm [23]. We initialize the solver at its previous output (or the zero configuration at initialization). We use the PyTorch Kinematics library to compute and differentiate through the forward kinematics function of the robot hand [24]. The retargeter runs at approximately 8 Hz when unconstrained and at 4.5 Hz with a constraint activated on an AMD Ryzen 3955WX 2.2 GHz CPU, which we found to be suitably reactive for teleoperated control. The computation time is dominated by the optimization routines, while the GP computation takes only 9 ms on average.

---

**Algorithm 1** Full computation of the desired joint angles

---

1: **function** COMPUTEDESIREDQ($\mathbf{q}_h$, $D$, $\boldsymbol{\theta}$, $\beta$, $\gamma$, $R_c$)
2:     Update $R_c$ from user input (pedal presses)
3:     Compute $\mathbf{q}_o^*(\mathbf{q}_h; \beta, \gamma)$ from Equation 1
4:     Allocate $\mathbf{q}_d$
5:     **for** $f \in F$ **do**
6:         Compute $\boldsymbol{\xi}^{f^*}(\mathbf{q}_h[f]; D, \boldsymbol{\theta})$
7:         $\mathbf{q}_d[f] \leftarrow \mathbf{a}(\boldsymbol{\xi}^{f^*} + \mathbf{V}(\mathbf{q}_o^*[f]))$                 $\triangleright$ $\mathbf{a}(\cdot)$ is the inverse of $\mathbf{V}(\cdot)$
8:     Compute $\mathbf{q}_c$ from Equation 5
9:     **return** $\mathbf{q}_c$

---

**Hardware and Control.** After the desired joint configuration is computed, it is sent to a lower level (gravity compensated) PD torque controller that tracks this set point at 300 Hz (smoothed by an exponential filter). This controller allows the operator to control the force each finger exerts on a grasped object by moving the set point for that finger into or out of the object. We use an Allegro right hand outfit with Xela sensors as our robot hand and use a Manus Quantum Metaglove to track

the operator's right hand's fingertip poses and joint angles. The Xela sensors are not used in our teleoperation interface, but do alter the coefficients of friction between the hand and target objects. We use an iKKEGOL USB triple foot pedal to allow the operator to toggle constraints on each finger.

## 5 Experimental Results

To evaluate our retargeter, we calibrated it to two different operators and measured their ability to complete 6 tasks requiring substantial dexterity from the robot's fingers, including several finger gaiting tasks. Calibration takes on average 4.5 minutes (including training the GP), making it easy to calibrate the retargeter to each operator individually.

### 5.1 End-to-End Testing

We evaluate our system on the 6 challenging tasks shown in Figure 1:

1. Horizontal Cube Rotation (Rot H): The operator must lift the cube off of the robot's palm and rotate it $180°$ about the horizontal axis before placing it back on the palm.

2. Vertical Cube Rotation (Rot V): The operator rotates the cube $90°$ about the vertical axis.

3. Screwdriver Palm to Quadpod (P-to-Q): The operator starts with a screwdriver on the robot's palm and transitions it to a "quadpod" grasp (suitable for turning the screwdriver).

4. Wrench Pickup (Wrench): The operator picks up a wrench off of a table by first pinching it, lifting it up, then transitioning to power grasping it.

5. Card Pickup (Card): The operator picks up a card off of a table by sliding it over the edge and pinching it between two fingers.

6. Screwdriver Controlled Pivot (Pivot): The operator grasps a screwdriver with all four fingers then loosens their grip until the screwdriver rotates to point downwards.

Compared to previous dexterous manipulation systems [7, 2, 4], we demonstrate our retargeter without any arm motion so that all dexterity must come from finger-level manipulation. The retargeter is kept the same across all tasks and attempts: no hand-tuning is performed for individual tasks. Tasks 1-4 require finger gaiting, while the grasps required by tasks 4-6 demonstrate our retargeter retains basic grasping functionality and precise control. Previous retargeters we have tried [7, 6] were unable to complete the finger gaiting tasks.

We had two operators (paper authors) attempt to complete all 6 tasks 5 times in a row, both with and without the use of the finger constraints (subsection 3.3). Operators were allowed to attempt the tasks several times before starting their scored runs to get accustomed to the retargeter. In runs with the finger constraints, operators were required to use them at least once. For each attempt, the operator earned 1 point for successful completion of the task, 0.5 points for reaching the desired final state but violating a task constraint (e.g., introducing rotation about unwanted axes during Rot H or Rot V, failing to control the screwdriver's pivoting in Pivot), and 0 points otherwise. We recorded each operator's completion times for attempts where they scored any points. Figure 4 shows the average scores achieved by each operator under each condition.

Both operators were able to complete several tasks with relatively high reliability given their difficulty, on average succeeding in 80% of trials (computed based on the best of the two conditions—in practice, the use of constraints is optional). Operators demonstrated advanced finger-gaited manipulation as well as common grasps (power and pinch grasps). Operators completed the tasks relatively quickly, taking on average only 43 and at most 139 s, indicating that the overall interface is fluent and usable. This is also fast enough for operators to collect a few hundred demonstrations in a few hours. Interestingly, the benefit of using finger constraints varies between tasks. Constraints are most effective in the Rot H, P-to-Q, Wrench, and Card tasks, where the operator must hold the target object steady as they use other fingers to adjust its pose, or use precise finger control to grasp a very thin object. However, in the Rot V and Pivot Tasks, the use of constraints sometimes reduces

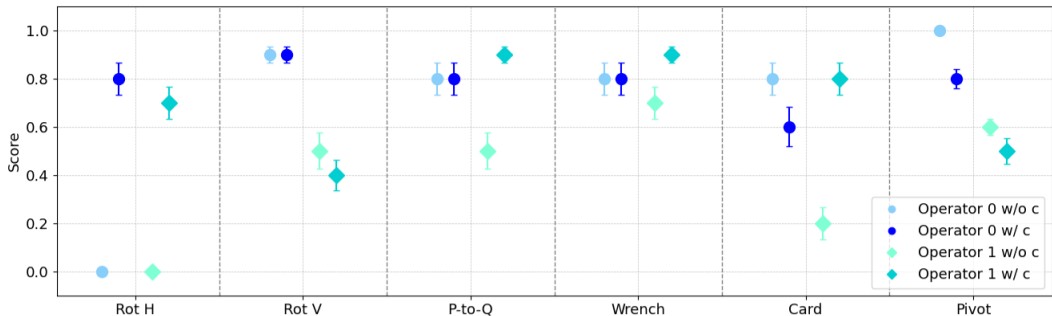

Figure 4: Task performance for two pilots on the 6 tasks with and without using the fingertip constraints. The plot shows the average scores; error bars show the relative variability ($0.2\sigma$).

Table 1: Reachable workspaces of each of the retargeters.

| Method | Joint Workspace ($\text{rad}^4$) | Fingertip Workspace ($\text{cm}^3$) |
|---|---|---|
| Joint | 0.1196 | 912.275 |
| IK | 0.1198 | 462.625 |
| HKVM | 0.2028 | 1090.750 |
| DexPilot [7] | 0.1577 | 908.950 |
| NN | 0.0627 | 500.325 |
| GP | 0.0718 | 706.600 |
| Res-NN | 0.1749 | 1170.450 |
| Res-GP (Ours) | **0.2399** | **1511.050** |

the operator's performance. In the case of the Pivot task, constraints simply serve as a distraction, since once activated, the operator has no control over the force being applied between the finger and thumb. In the Rot V task, once the constraint was activated, it was difficult to command the fingers in a proper arcing motion to achieve the desired rotation. This constraint is better suited to maintaining an existing grasp of an object and translating it.

## 5.2 Comparison With Previous Methods

We hypothesize that this retargeter's ability to complete these difficult tasks stems from its combination of an enlarged reachable workspace and preservation of precise operation in regions that require it (for example, when pinching the fingers). To test this hypothesis, we recorded a video of a hand moving through all regions of its workspace and collected a long trajectory of $N$ human hand joint angles by having an operator attempt to mimic the hand in this video several times in a row. We then used several baseline retargeters to compute a set of robot joint trajectories $(\mathbf{q}_{r_i}^j)_{i \in [N]}$ where $j$ denotes which retargeter was used. To approximate the robot hand's reachable workspace under a retargeter, we considered each robot finger individually and voxelized its joint space at resolution $\delta_1 = 0.05$ rad. We report the volume of the voxels traversed by the retargeted robot trajectory, summed across all fingers. To compute the fingertip workspace, we performed the same procedure with each fingertip's trajectory, using a spatial resolution of $\delta_2 = 5$ mm, tracking only the fingertip's position (disregarding orientation). We performed this procedure with 5 operators (calibrating the relevant retargeters to each operator) and report the average results.

Table 1 lists the retargeters we tested and their joint and fingertip workspaces. The "Joint" retargeter linearly rescales each of the 16 tracked human finger joint positions to match the range of each robot joint. The inverse kinematics ("IK") retargeter uses IK to solve for a joint configuration of each robot finger that places the fingertip at the same position as the corresponding human finger relative to the robot palm. HKVM directly uses $\mathbf{q}_o^*$ as the desired joint target, and DexPilot adds a finger-snapping and collision avoidance heuristic to HKVM [7]. The neural network ("NN") and Gaussian Process ("GP") retargeters learn to output a desired joint configuration from a human finger configuration directly from the calibration data. Finally, the residual neural network ("Res-NN") uses a neural network, rather than a GP, to learn a residual on top of HKVM.

Res-GP achieves by far the highest joint and fingertip workspaces of any retargeter. Even though the Joint retargeter can reach the full range of each robot joint independently, its workspace suffers from failing to consider correlated movements of human fingers. The fingertip retargeting methods (IK, HKVM, and DexPilot) are mostly concerned with configurations in the vicinity of fingertip grasps, and tend to severely limit the fingers' and thumb's mobility near the base of the palm. The other learning-based methods (GP, NN, and Res-NN) tend to overfit to the calibration data, "snapping" quickly between calibration poses rather than smoothly interpolating between them. Res-GP in contrast, uses a strong prior of how to interpolate between fingertip configurations (encoded by HKVM as a base retargeter and our choice of kernel parameterization), allowing it to both reach the far extents of the workspace (specified by the calibration poses) while allowing for smooth control in its interior. Additionally, while drastically expanding the robot's workspace, Res-GP still allows the operator to precisely control the fingers in critical regions like fingertip pinches. Figure 5 shows how the Joint retargeter does a poor job of approximating this pinch configuration, and while the HKVM retargeter can approximate the pinch, the location of the pinch remains nearly static relative to the palm. In contrast, Res-GP is able to pinch both near and far from the palm. This shows how Res-GP effectively captures where it can trade off sensitivity for reachability.

## 6 Limitations

Our work has several limitations. First, we rely on calibration poses tailored to a specific robot hand. This assumes that operators will have time to collect calibration data before beginning operation. While empirically this can be performed quickly and only needs to be done once for each operator, it introduces an additional step compared to calibration-free retargeters [7, 2, 4]. Additionally, we rely on intuition of the required robot hand workspace to determine the set of calibration poses. We provide a heuristic for choosing these poses that could be applied to other hands, but a systematic approach to identifying them would make this method more automated.

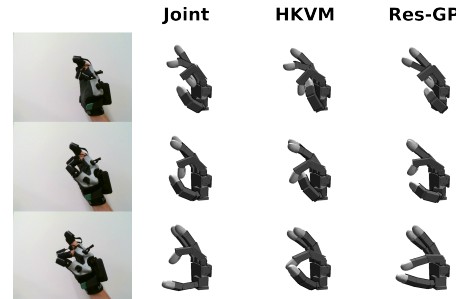

Figure 5: Qualitative comparisons between retargeters. Res-GP reproduces the operator's pinch at many distances from the palm.

Second, while our finger constraints enable the operator to maintain stable contact with a grasped object while translating it, they don't allow for intuitive adjustment of the applied force, and rotating the constrained fingers about a center point proved difficult. Additionally, the operator is currently required to use a separate interface (foot pedals) to toggle the constraints, rather than only using their hand to control the robot. In the future, we'd like to address this by modifying how constraints are applied to allow the operator to more intuitively move objects in their grasp, using an approach similar to that presented in [25]. We'd also like to investigate how the operator's intent to use a constraint can be inferred directly from their hand tracking data.

Finally, as our method relies on optimization routines, the retargeter can get trapped in local minima and stop tracking the operator. We rarely encounter this problem in our experiments and typically the operator is able to recover from it, but a more robust solution is needed for real deployments.

## 7 Conclusion

We present a retargeter for finger-gaited dexterous manipulation with multi-fingered robot hands. Our method learns a residual GP on a small set of calibration poses to enhance the robot's reachable workspace, and allows the operator to constrain fingertip motions to maintain stable contacts. Based on real-world experiments, our method is fast to calibrate and use, and enables previously unseen levels of teleoperated dexterity. In future work, we'd like to investigate how calibration poses can be chosen in a more systematic way, and how the raw retargeting and constraint satisfaction steps can be consolidated while maintaining the same level of dexterity.

**Acknowledgments**

We'd like to thank Benjamin Evans for thoughtful comments on the paper and naming the method.

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

# A  Training and Implementation Details

Algorithm 2 describes how the GP for each finger is trained.

---

**Algorithm 2** Training the GP

---

1: **function** TRAINGP($D$, $\alpha$, $E$, $f$)
2:     Initialize $\boldsymbol{\theta}$ randomly
3:     Vectorize the sequence of $\boldsymbol{\xi}_i[f]$ in $D$ into $\mathbf{Q}_r$
4:     **for** $e \in [E]$ **do**
5:         Compute $\mathbf{K}$ for all pairs of $(\mathbf{q}_{h_i}[f], \mathbf{q}_{h_j}[f]) \in D$ using Equation 3
6:         $\mathbf{K}_\sigma \leftarrow \mathbf{K} + \sigma^2 I$
7:         $\mathbf{K}_q \leftarrow \mathbf{K}_t \otimes \mathbf{K}_\sigma$ using the current value of $\mathbf{K}_t$
8:         Compute loss $\mathcal{L}$ from Equation 4
9:         Compute $\nabla_{\boldsymbol{\theta}}\mathcal{L}$ and update $\boldsymbol{\theta}$ using Adam [21] with learning rate $\alpha$
10:     **return** $\boldsymbol{\theta}$

---

In our implementation, we set the number of epochs $E = 3000$ and the learning rate $\alpha = 0.01$, which we found experimentally to be sufficient to reach convergence. The full calibration procedure takes on average (across 5 tested operators) 4.5 minutes. Data processing and parameter fitting for the GP takes a total of 65 seconds, while explaining the calibration procedure and collecting the calibration data takes 3 minutes and 20 seconds.

During inference, we need to compute the mean $\boldsymbol{\xi}^{f*}$ of the posterior distribution of the residual conditioned on the observed calibration poses. If $C_f$ datapoints $((\mathbf{q}_{h_i}[f]), \boldsymbol{\xi}_i^f))_{i\in[C_f]}$ have already been observed for finger $f$ with additive independent and identically distributed Gaussian noise with variance $\sigma^2$, the conditional distribution of $\boldsymbol{\xi}^f(\mathbf{q}_h^*[f])$ at a new input $\mathbf{q}_h^*[f]$ can be computed as

$$\boldsymbol{\xi}^f(\mathbf{q}_h^*[f]) \sim \mathcal{N}(\mathbf{k}^\intercal \mathbf{K}_\sigma^{-1} \bar{\mathbf{q}}_h, k(\mathbf{q}_h^*[f], \mathbf{q}_h^*[f]) - \mathbf{k}^\intercal \mathbf{K}_\sigma^{-1} \mathbf{k}) \tag{6}$$

where

$$\mathbf{K}_\sigma = \begin{bmatrix} k(\mathbf{q}_{h_1}[f], \mathbf{q}_{h_1}[f]) & \cdots & k(\mathbf{q}_{h_1}[f], \mathbf{q}_{h_{C_f}}[f]) \\ \vdots & \ddots & \vdots \\ k(\mathbf{q}_{h_{C_f}}[f], \mathbf{q}_{h_1}[f]) & \cdots & k(\mathbf{q}_{h_{C_f}}[f], \mathbf{q}_{h_{C_f}}[f]) \end{bmatrix} + \sigma^2 \mathbf{I}$$

$$\mathbf{k} = \begin{bmatrix} k(\mathbf{q}_{h_1}[f], \mathbf{q}_h^*[f]) & \cdots & k(\mathbf{q}_{h_{C_f}}[f], \mathbf{q}_h^*[f]) \end{bmatrix}^\intercal$$

$$\bar{\mathbf{q}}_h = \begin{bmatrix} \boldsymbol{\xi}_1^f & \cdots & \boldsymbol{\xi}_{C_f}^f \end{bmatrix}^\intercal.$$

We use this expression to compute the posterior mean when using the retargeter in Algorithm 1.

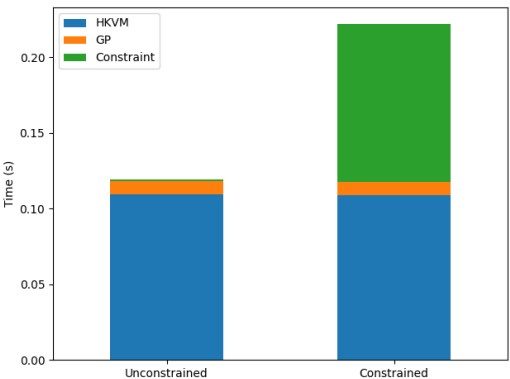

Figure 6: Time to run Res-GP for a trajectory where $R_c$ is empty (Unconstrained) and one where it contains the index-to-thumb vector (Constrained). The GP incurs little overhead.

When applying the constraint specified by Equation 5, we found that hand-tuning $d$ to 1 cm applied sufficient force to our target objects to lift and manipulate them. As shown in Figure 2, the keypoints of each finger are located inside the fingertip geometry, so that constraining these points to be 1 cm away from each other causes the fingertips to intersect one another. This allows the hand to stably grasp very thin objects as well. In our implementation, we restricted $R_c$ to only contain vectors from fingertips to the thumb tip so that we could easily control membership in $R_c$ with foot pedals. With a more extensive interaction interface, $R_c$ could conceivably contain other keypoint vectors, for example, between the index and middle fingers, to grasp objects in different ways.

Figure 6 shows a breakdown of the time taken by different computations during inference. With no constraints activated the retargeter runs at approximately 8 Hz; with the index-to-thumb vector constrained, the retargeter runs at approximately 4.5 Hz. The optimization for the HKVM base retargeter dominates the computation time, while the GP inference is quite fast, taking about 9 ms on average (performing inference for all four fingers takes a total of 9 ms). This shows that our residual GP can be added onto many different base retargeters with little additional overhead. We use NLopt to limit the optimization time in HKVM to 100 ms, and it uses up this entire time budget. When no constraints are activated, the constraint satisfaction problem has an overhead of approximately 1 ms, but when a constraint is activated, it takes on the order of 100 ms to solve. As shown by our experiments, we found that this high-level control rate sufficed for quasistatic and quasidynamic manipulation tasks. To handle highly dynamic tasks, a higher control rate and more advanced feedback are likely required.

We use an Allegro right hand outfit with Xela sensors as our robot hand and use a Manus Quantum Metaglove to track the operator's right hand's fingertip poses and joint angles. We use an exponential filter to smooth the values of $\mathbf{q}_c$ commanded by the operator before using a PD torque controller to track this smoothed target on the robot hand at 300 Hz. The smoothed target joint configuration at time $t$ is computed as $\mathbf{q}_t^{\text{target}} = a \cdot \mathbf{q}_c + (1 - a) \cdot \mathbf{q}_{t-1}^{\text{target}}$, where $\mathbf{q}_0^{\text{target}}$ is initialized to 0. We hand-tuned $a = 0.01$ to achieve smooth tracking. When tracking the operator's hand, we remove Manus' "pinch correction," since we found that it introduced tracking artifacts, detecting that the fingers and thumb were performing a pinch even when they were quite far from one another. We prioritized stability and consistency of the hand tracking over absolute accuracy. We additionally opted not to use Manus' built-in glove calibration since we found that it visibly decreased the tracking accuracy of the glove.

## B  Baselines and Evaluation Metrics

Here, we provide more complete descriptions of the baseline retargeters tested:

- Joint Space (Joint): We remap each human joint to the corresponding robot joint (ordered by proximity to the palm). For each human joint, we find its limits (from the recorded workspace) and linearly rescale its range to match the robot's joint range. To run the retargeter, we apply these scalings to each of the human's current joint values to find the desired robot joint configuration.

- Inverse Kinematics (IK): We track each of the human's fingertips in their palm frame and solve an IK problem for each of the robot's fingers to place the corresponding fingertip at that position (ignoring orientation). The palm of the robot is placed as shown in Figure 2.

- Hand Keypoint Vector Matching (HKVM): We directly use $\mathbf{q}_o^*$ (from Equation 1) as the desired joint configuration.

- DexPilot [7] (DexPilot): Essentially the same method as HKVM but using the heuristics presented in Handa et al. [7] for precise fingertip pinching and collision avoidance.

- Neural Network (NN): We train a neural network for each finger on the calibration data to directly output the finger's target joint configuration based on the operator's finger configuration. We use the same rotation representation for the NN retargeter that we used for the GP. The network is a fully-connected multi-layer perceptron with an input size of 8,

4 hidden layers of 256 neurons each, and an output layer of size $2|f|$, which we interpret as a $|f| \times 2$ matrix, each row specifying a joint angle as a vector. We use rectified-linear non-linearities after each of the hidden layers. The network is trained for 5000 epochs using the AdamW optimizer with learning rate 0.001 (stepped down to 0.0001 after 2000 epochs), which we found to yield perfect loss on the training set. We use the negative mean cosine similarity loss between each of the rows of the network output and the $\mathbf{V}$ matrix representing the ground-truth Allegro hand joint position at each calibration pose.

- Gaussian Process (GP): We directly train a GP for each finger on the calibration data to output the finger's target joint configuration. We use a constant mean function and the same kernel and rotation representation presented here.

- Residual Neural Network (Res-NN): Use a neural network to learn a residual on top of HKVM. We learn a separate network for each Allegro hand finger. Each network has the same architecture and is trained in the same way as the NN retargeter, except its output is summed with the HKVM output (for a given finger) to produce its prediction for each human hand configuration.

- Residual Gaussian Process (Res-GP): The method presented here.

We found that the two operators whose results are presented in the paper were not able to complete any of the finger gaiting (Rot H, Rot V, P-to-Q, and Wrench) tasks with the baseline retargeters.

**Workspace Analysis.** To compute the reachable workspace of each of the retargeters, we collected a long trajectory of $N$ human hand joint angles by having an operator attempt to mimic a video of a hand moving through its workspace several times in a row. We use this approach to approximate the operator's hand workspace, rather than sampling on a kinematic model of the operator's hand, to capture subtleties such as configuration-dependent joint limits of human fingers. We then processed this trajectory using each retargeter to generate trajectories of robot joint angles and used Algorithm 3 to approximate the reachable workspace of the retargeter, considering each finger independently and discretizing the robot's joint space into hypercubes of size $\delta$. $|f|$ denotes the number of joints associated with the robot finger $f$ (in the case of the Allegro hand, always equal to 4). We then sum the workspace of each finger to arrive at the retargeter's total reachable workspace. To compute the workspace of the robot's fingertips under each retargeter, we similarly run Algorithm 3 but replace Line 6 with $X \leftarrow X \cup \lfloor p(\mathbf{q}_{r_i}^j[f])/\delta \rfloor$ and Line 7 with $w \leftarrow w + |X|\delta^3$. Here, $p(\cdot)$ gets the position of the relevant robot fingertip in its palm frame.

---

**Algorithm 3** Compute joint workspace

1: **function** COMPUTEJOINTWORKSPACE($(\mathbf{q}_{r_i}^j)_{i \in [N]}, \delta$)
2:     $w \leftarrow 0$
3:     **for** $f \in F$ **do**
4:         $X \leftarrow \emptyset$
5:         **for** $i \in [N]$ **do**
6:             $X \leftarrow X \cup \lfloor \mathbf{q}_{r_i}^j[f]/\delta \rfloor$         $\triangleright \lfloor \cdot \rfloor$ and division applied elementwise
7:         $w \leftarrow w + |X|\delta^{|f|}$
8:     **return** $w$

---

We performed this procedure for five different operators with hand widths (thumb tip to pinky tip) ranging from 18.0 to 22.5 cm and heights (base of palm to tip of middle finger) ranging from 16.5 to 18.5 cm. Table 2 shows the joint (J) and fingertip (F) workspaces for each of the subjects in $\text{rad}^4$ and $\text{cm}^3$ respectively. We see that not only is the average (across operators) workspace of Res-GP superior to the baseline retargeters, it is higher for every individual operator as well.

**Qualitative Comparisons.** Figure 7 shows the robot configurations produced by all of the tested retargeters for the operator hand configurations shown in Figure 5, as well as for extreme positions of the thumb. The IK retargeter tends to keep the robot's fingers quite close to the palm since no scaling

Table 2: Reachable workspaces of each of the baseline retargeters for all 5 tested operators. Joint (J) and fingertip (F) workspaces for each of the subjects are reported in rad$^4$ and cm$^3$ respectively.

| Method | Subject 0 J | Subject 0 F | Subject 1 J | Subject 1 F | Subject 2 J | Subject 2 F | Subject 3 J | Subject 3 F | Subject 4 J | Subject 4 F |
|---|---|---|---|---|---|---|---|---|---|---|
| Joint | 0.124 | 930 | 0.122 | 959 | 0.123 | 865 | 0.107 | 857 | 0.123 | 951 |
| IK | 0.120 | 422 | 0.123 | 446 | 0.113 | 395 | 0.108 | 526 | 0.135 | 524 |
| HKVM | 0.189 | 1000 | 0.200 | 998 | 0.197 | 1009 | 0.185 | 1140 | 0.244 | 1307 |
| DexPilot [7] | 0.145 | 842 | 0.150 | 811 | 0.150 | 824 | 0.151 | 966 | 0.192 | 1101 |
| NN | 0.080 | 632 | 0.072 | 582 | 0.064 | 506 | 0.034 | 310 | 0.064 | 472 |
| GP | 0.074 | 715 | 0.077 | 756 | 0.062 | 566 | 0.066 | 682 | 0.080 | 813 |
| Res-NN | 0.159 | 1106 | 0.173 | 1230 | 0.153 | 976 | 0.166 | 1174 | 0.224 | 1367 |
| Res-GP (Ours) | **0.241** | **1645** | **0.231** | **1407** | **0.236** | **1355** | **0.209** | **1457** | **0.282** | **1691** |

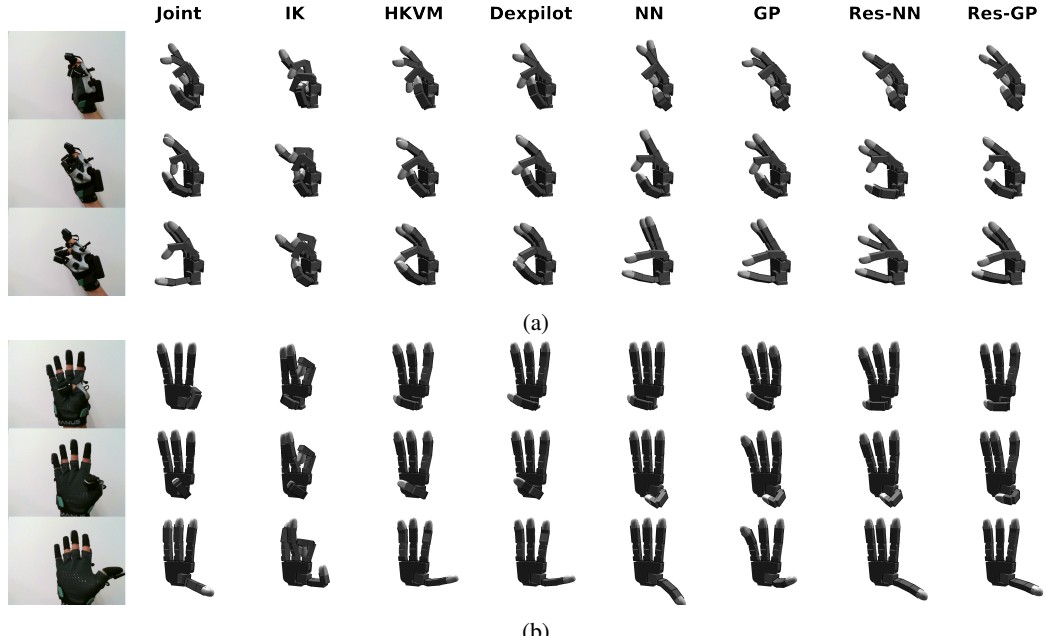

(a)

(b)

Figure 7: Qualitative comparison between retargeting methods for a series of human hand configurations.

is applied to the operator's hand, and often gets trapped in local minima. DexPilot behaves quite similarly to HKVM in these regions. The learning based methods all produce similar results because the demonstrated human hand configurations are very similar to configurations in the calibration set (Figure 3). However, the NN, GP, and Res-NN retargeters tend to overfit to the calibration data, resulting in much lower reachable workspaces.

The higher workspace of Res-GP allows it to reach many parts of target objects, giving the operator many choices of contact location, and ultimately enabling advanced finger gaiting. In particular, it allows the operator to place the robot's thumb in many locations, as shown in Figure 7. The operator is able to move the thumb all the way to the base of the palm, which none of the uncalibrated retargeters can achieve. Additionally, using Res-GP, the operator is able to keep the thumb outside the space above the palm where an object might sit, whereas other retargeters, such as Joint and HKVM, substantially intrude into this region during the demonstrated trajectory. When manipulating an object, this is likely to disturb it and introduce undesired motion. Res-GP gives the operator more freedom to reposition the thumb without touching the target object until they wish to.

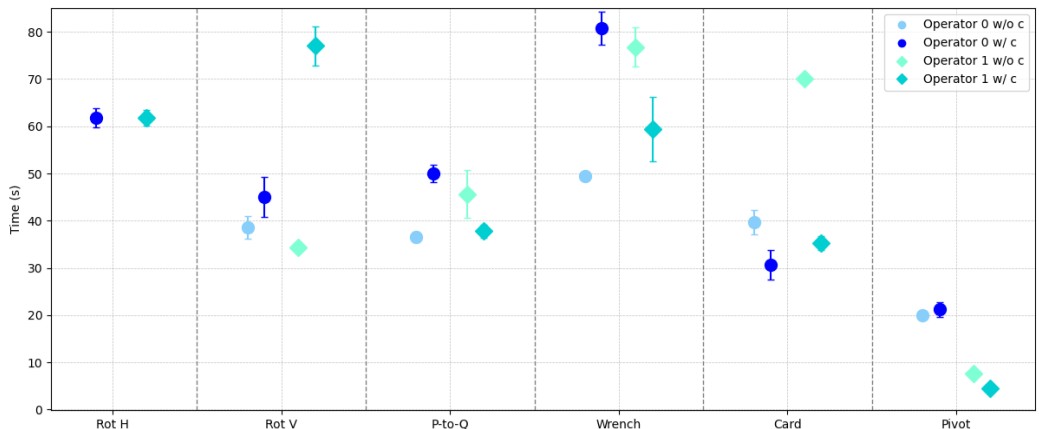

Figure 8: Times taken by each operator under each condition to perform the tasks. The plot shows the average scores; error bars show the relative variability ($0.2\sigma$). Operators take an average of about 43 seconds to collect each successful demonstration.

**Timing Results.** Figure 8 shows the mean and relative variation of the time taken for each operator to complete each task across their 5 attempts both with and without the constraints. Times for all tasks are reasonable, typically less than a minute, opening the potential for fast collection of many demonstrations for each task. The operators respond quite differently to the constraints. Operator 0 appears to be less comfortable with the constraints, taking longer to complete most tasks when using them. Operator 1, in contrast, fluidly integrates them into their operation, likely allowing them to more confidently complete each task and thereby reach the desired final state more quickly.

