# OpenReview forum: "ResPilot: Teleoperated Finger Gaiting via Gaussian Process Residual Learning"
_robot-learning.org/CoRL/2024/Conference — CoRL 2024_

### Official Review · Reviewer_5i54 · 2024-07-19
**A promising teleoperation system for dexterous hand manipulation**

**Originality:** 5
**Technical Quality:** 4
**Clarity Of Presentation:** 4
**Potential Impact:** 3
**Recommendation:** 4
**Confidence:** 2

**Review:**

Strengths:
- The paper is clearly written with enough technical details for reproducibility.
- The paper is positioned very well in the literature. The reviewer can clearly understand the differences and significance as compared to other works.
- The proposed method about using a residual GP combined with keypoint mapping is interesting and promising.
- There are enough details written in the limitation section, which help the readers understand better about the underlying assumptions.
- The reviewer appreciates the efforts on having the supplementary video include all the recorded real-robot experiments.

Weaknesses:
- Overall the experiment results on real hardware look extensive and convincing, though the reviewer still hopes that there are at least three operators that can be involved in the whole evaluation process.
- If the reviewer understands correctly, there is no contribution coming from this paper that is included in Section 3.1. If that is the case, it should be removed from the method section and included in background or preliminary.
- (minor) Fix typo for references in line 199.

**Quality Of The Limitations Section:**

3

**Questions For Rebuttal:**

- Can the authors conduct an additional set of experiments with another operator?
- The reviewer noticed failure cases in the video, but is still curious about what's the main reason behind those failures. Can the authors elaborate on this?

**Robotics Focus:**

4

**Summary Of Paper:**

This paper focuses on teleoperation for dexterous hand manipulation. The authors introduce a novel retargeting method for mapping the human hand to the robot hand's configuration using Gaussian process. By evaluating the teleopration system through six challenging tasks, it succeeds in an average of about 80% of the time. There are also quantitative evaluations showing their method's superiority against baselines from the litrature.

**Summary Of Recommendation:**

The reviewer believes the proposed method has made significance progress compared against the state of the art, and potentially opens up new research opportunities for dexterous hand manipulation.

---

### Official Review · Reviewer_zYVG · 2024-07-22
**Review before Rebuttal**

**Originality:** 4
**Technical Quality:** 3
**Clarity Of Presentation:** 4
**Potential Impact:** 3
**Recommendation:** 3
**Confidence:** 4

**Review:**

This paper presents a novel approach to teleoperated finger-gaited manipulation using multi-fingered robot hands. The methodology combines optimization-based hand keypoint vector matching with learned residuals via GP, expanding the robot's workspace and enabling complex manipulations.

The work's primary strength lies in its innovative combination of existing techniques to address specific challenges in dexterous teleoperation. The method enables more complex finger-gaited manipulations than previously demonstrated, showing potential for skill transfer and data collection for autonomous systems. The evaluation of six challenging tasks demonstrates the system's capabilities effectively.

However, the paper has some limitations.
1. The experimental evaluation lacks comprehensive comparisons with other methods, especially the calibration-free method mentioned in the related work. The current Table 1 is great for comparison but no task information is involved in this comparison, e.g. success rate, time consumption of teleoperation. More analytical results are necessary.
2. Another bottleneck is the retargeting speed. The author claims that the current 4.5 Hz speed is suitably reactive for teleoperated control, which is not plausible to me. To perform in-hand manipulation tasks, a much higher control frequency is necessary, the 5Hz speed is very laggy for human operators so the operator needs much training before they can use the teleoperation system.
3. Additionally, while the potential for downstream applications is significant, this aspect is not fully explored in the current work, which is a tiny weakness of this paper.

**Quality Of The Limitations Section:**

3

**Questions For Rebuttal:**

To better present this work, it is better to provide the following during rebuttal:
1. A more comprehensive analysis of the system latency, and ways to improve the retargeting speed to a reasonable value.
2. More experiments to compare with existing work, e.g. [7] and other calibration-free retargeters.
3. Visualization and analysis of the calibrated robot pose, and the corresponding robot pose generated by other calibration-free retargeters using the same human pose.

**Robotics Focus:**

4

**Summary Of Paper:**

This paper presents a novel teleoperation method for dexterous robot hands, enabling complex finger-gaited manipulation. Key innovations include: 1. A retargeting approach combining optimization-based hand keypoint matching with learned residuals via Gaussian Processes, expanding the robot's workspace. 2. Fingertip distance constraints to maintain stable object contacts.

**Summary Of Recommendation:**

The problem investigated by this paper is important, but the performance of the system, as well as the limitation in experiments make this paper not ready to be accepted.

---

### Official Review · Reviewer_MZs1 · 2024-07-26
**ResPilot Review**

**Originality:** 4
**Technical Quality:** 4
**Clarity Of Presentation:** 5
**Potential Impact:** 3
**Recommendation:** 3
**Confidence:** 4

**Review:**

The method proposed in ResPilot utilizes a residual gaussian process used to model the residual control to retarget a human hand to a four-fingered from a base-retargeter DexPilot which used human keypoint-vector matching, proposed by Handa et al. [7]. The model is fit by using a set of ground truth labeled pairs between human and robot matching poses, and then fits a separate residual GP model to each finger, taking the human finger as input.  An additional constraint model is also imposed on top of the residual model to constrain the solutions of different fingertips to maintain a distance to the thumb tip.

Overall, the key strengths of this paper is in
a) the simplicity of training the model learned to improve hand-retargeting, requiring only 24 hand-labeled examples
b) the qualitative and quantitative improvements upon the previous retargeting baseline, showing flexible calibration across two operators, and difficult tasks that would require complex dexterous control.

Some suggestions/weaknesses to address:
a) Include an ablation of the method without using footpedal constraints, but using the same constraints that are used in DexPilot (if this is what is in figure 4, please correct me as it is unclear whether that is the case).
b) The details on the controller speed/frequencies should be moved to the main body, in addition to the appendix, as those are important details for a potential user who may be determining to use this approach for teleoperation.

**Quality Of The Limitations Section:**

3

**Questions For Rebuttal:**

Overall the paper was very well written, and clearly presented. Some remaining questions after reading through it and going through the supplemental materials include:

- Does including this additional foot-pedal controller make the comparisons to the other methods fair when the workspace can be obviously expanded with additional degrees of freedom/controls for the operator?
- Some of the tasks do not seem to clearly show the emphasized finger-gaiting, such as Pivot, which is more of a slip task. Were tasks such as rotating the screw driver once grasped possible for the operators (Task 3)?
- There is no direct comparison between methods DexPilot and ResPilot on any of these tasks. Were these newly selected tasks simply not possible without residual GPs? If so, a note of this would be useful to further motivate.

**Robotics Focus:**

4

**Summary Of Paper:**

This paper proposes a optimization-based method for teleoperation of dexterous hands to enable finger-gaiting and other complex dexterous hand motions with multi-fingered robot hands.

**Summary Of Recommendation:**

I think this paper yields an interesting and novel approach to improve pose retargeting for dexterous teleop, although some of the motivating tasks and examples do not fully convey this, nor are there comparisons to prior methods to strongly justify the additional complexity. However the results shown are fairly clearly, and would upgrade to a strong accept if these points are clarified/addressed.

---

### Author Rebuttal · Authors · 2024-08-05

We thank the reviewers for their insightful comments. Multiple reviewers have mentioned a limitation of our experiments in that we do not perform any end-to-end comparisons of our method against DexPilot. While it is true that we did not perform a systematic experiment to verify the capability of DexPilot, both operators found completing several of the presented tasks impossible with DexPilot. We have added a note to section 5.1 remarking on the difficulty of performing finger gaiting with previous retargeters. Furthermore, we show that DexPilot’s reachable workspace is significantly limited compared to ResPilot’s, making some configurations required in certain tasks simply unreachable. For example, in Tasks 2 and 3, the Allegro thumb needs to move to the base of the palm, which is not possible for DexPilot. Additionally, looking at the tasks the original DexPilot authors chose, all but one of them are grasping or pushing tasks from the perspective of the hand. The last task was a brick rotation task, similar to our Horizontal Rotation task, but much of the motion used to complete this task is generated from the robot’s arm, not the hand itself. We'd argue that the tasks demonstrated in ResPilot are in an entirely different category of difficulty than the capabilities shown by DexPilot (finger gaiting vs. grasping and pushing). As such, even without a large-scale human subjects study to precisely quantify the performance gap between the two retargeters, we argue that comparing our current results to those presented by DexPilot makes it highly likely that this gap is significant.

We have responded to other specific concerns in “Official Comments” for each review. We additionally uploaded an updated manuscript reflecting the changes in our responses.

---

### Decision · Program_Chairs · 2024-09-04

**Decision:**

Accept

**Comment:**

The paper's strengths include:
The paper introduces a novel method for the teleoperation of dexterous hands by combining optimization-based keypoint matching with learned residuals via Gaussian Processes, improving complex finger-gaiting and manipulations.
The method shows promising results in both qualitative and quantitative evaluations, demonstrating improvements over baseline methods and effective performance in challenging tasks.
The paper is well-written and presented, with sufficient technical details for reproducibility and a thorough explanation of the proposed method.
The system is tested on real hardware with a detailed video showing experiments, supporting the practical applicability of the approach.

The paper's weaknesses include:
There is insufficient comparison with existing methods, especially calibration-free retargeting methods. The paper should provide more analysis of how the proposed method stands relative to others.
The system’s retargeting speed (4.5 Hz) is questioned as being potentially too slow for effective teleoperation, particularly for in-hand manipulation tasks.
The paper lacks experiments with multiple operators and a detailed analysis of failure cases. Additionally, certain experimental details, such as controller speed and constraints, need to be moved to the main body for clarity.
Some tasks in the evaluation do not clearly demonstrate the emphasized finger-gaiting, and there is no direct comparison of the performance of the new method with the baseline DexPilot on all tasks.

Post-rebuttal meta-review:
The paper addresses the challenge of teleoperating dexterous robot hands for complex manipulation tasks. Current methods struggle to replicate the full functional workspace of the human hand, particularly for intricate tasks like finger-gaiting.

The authors propose ResPilot, a novel method that enhances teleoperated dexterous manipulation by expanding the robot hand’s reachable workspace. This is achieved through Gaussian Process (GP) residual learning, which calibrates a human hand to a robot hand motion retargeter. The system introduces fingertip distance constraints to maintain stable object contacts, allowing for more precise and flexible manipulation.

The method is innovative, combining optimization-based keypoint matching with learned residuals via Gaussian Processes to enable complex finger-gaiting tasks. The approach is well-positioned within the existing literature, and the paper is well-written, with sufficient technical detail to ensure reproducibility. The method demonstrates significant improvements over baseline retargeting approaches and is validated through extensive quantitative evaluations on challenging tasks.

The paper lacks a comprehensive comparison with other existing methods, particularly calibration-free retargeting methods. The retargeting speed (4.5 Hz) may be too slow for effective real-time teleoperation, particularly for in-hand manipulation tasks, raising concerns about its practicality. The paper would benefit from additional experiments involving multiple operators to assess generalizability, and a more detailed analysis of failure cases is needed to clarify system limitations.

Final Recommendation: Strong Accept --The paper presents a promising approach to teleoperated dexterous manipulation with clear improvements over existing methods. However, the concerns about retargeting speed, the need for broader comparisons, and further operator testing must be addressed to strengthen the paper’s contributions.